# Exploration of Cucumber Waste as a Potential Biorefinery Feedstock

Yang Gao [1,*] , Hannah K. Briers [2], Avtar S. Matharu [2] and Jiajun Fan [2,*]

1 State Key Laboratory of Separation Membrane and Membrane Process, School of Chemistry, Tiangong University, Tianjin 300387, China
2 Green Chemistry Centre of Excellence, Department of Chemistry, University of York, York YO10 5DD, UK
* Correspondence: yg1042@tiangong.edu.cn (Y.G.); alice.fan@york.ac.uk (J.F.)

**Abstract:** The exploration of cucumber waste as a potential biorefinery feedstock is reported. Initially, extractives (essential oils) were isolated from cucumber waste via vacuum microwave hydro-distillation (VMHD). The yield and quality of the extractive were compared with respect to traditional hydro-distillation (HD). The essential oils were obtained over a range of microwave power (500, 750, 1000 W) and vacuum pressures (100, 200, 300 mbar). The highest quality (0.49 wt %) was obtained at a microwave irradiation power of 500 W and a vacuum of 300 mbar. VMHD is much quicker and more energy-efficient than HD. Within the context of a zero-waste biorefinery, the extractive-free residues were the solid residues from two different extraction methods were compared and characterized by ATR-IR, $^{13}$C solid-state NMR spectroscopy, SEM, TGA, and CHN elemental analysis. The resultant residues are cellulosic-rich, and no significant changes were observed with VMHD and HD treatment. The results indicated that the utilization of these residues can provide an efficient, inexpensive, and environment-friendly platform for the production of cellulosic materials.

**Keywords:** cucumber; vacuum microwave hydro-distillation; hydro-distillation; cellulosic materials; essential oils; biorefinery

## 1. Introduction

Cucumber (*Cucumis sativus*), initially derived from southern Asia, is cultivated by humans with around 5000 years of historical records [1]. As a member of the Cucurbitaceae family, the cucumber is widely cultivated in temperate regions all over the world [2]. In 2020, cucumber production (*Cucumis sativus* L.) was approximately 9.13 million tons [3]. Cucumbers are typically processed for fresh products such as salad, pickles, or vegetable juice due to their high water content and good contents of vitamins and minerals [4]. Meanwhile, they can also be processed to extract value-added products such as essential oils. Cucumbers are naturally rich in essential oils which contain specific chemical constituents including 2,6-nonadienal, 2-nonenal, 3-nonenol, and 3-nonenal [5]; these complex mixtures of volatile organic compounds have potential multipurpose functional benefits in different fields such as agents and additives in food, perfumes, and especially pharmaceutical and biomedical industries due to their physicochemical properties such as antibacterial, antifungal, hypoglycemic, hypolipidemic, cytotoxic, and wound healing activities [6,7].

Extraction is a crucial step in utilizing the essential oils from cucumber, and cold pressing is the main isolation method available; nevertheless, several demerits restrict the development of this technique: low yield becomes an issue due to the incomplete removal of the oil content [8]. Recent extensive numerical studies have been considered on faster, more efficient, and cost-effective approaches to extract essential oils, including supercritical fluid extraction [9], ultrasound extraction [10], and subcritical water extraction [11].

Among these advanced techniques, microwave technologies are superior to conventional hydro-distillation due to their superior quantity of oils in minimum extraction time,

high extraction efficiency with less energy consumption, and higher yield [12,13]. During the process, microwave irradiation with a 2.45 GHz frequency activates the polar molecules of biomass, including water, and the whole solvent–matrix mixtures are heated homogeneously. In contrast, heating occurs only on the surface in traditional methods [14]. The cellular structure of the plant sample is split under microwave heating; the volatile compounds release and evaporate with water steam and then are distilled and collected by a cooling system. The technology can process the biomass directly without any pretreatment, and the heating is rapid and homogeneous; this makes the procedure adaptable for continuous processes and easily scalable [15]. Thus, various feedstocks including nutmeg [16], lemongrass [17], cinnamon [18], mint [13], and orange [15,19] were successfully applied. Therefore, the past decade has witnessed a pool of extraction techniques based on microwave energy, for instance, microwave-assisted hydro-distillation (MAHD), microwave-accelerated steam distillation (MASD), microwave hydro-diffusion and gravity (MHG), microwave steam distillation (MSD), solvent-free microwave extraction (SFME), and vacuum microwave hydro-distillation (VMHD) [20]. VMHD is an effective method for the extraction of compounds which is heat- and oxygen-sensitive [21], at the same time, given the low boiling point, the microwave energy consumption is lower than that of the normal microwave method.

However, to the best of our knowledge, there are very few reports of valorization of spent cucumber waste after oil extraction. It is estimated that 17% (by weight) of cucumber waste is produced after processing annually; the re-utilization is limited [22]. As the transformation of biomass and its waste to high-value chemicals could moderate environmental burdens [23], it is necessary to characterize the spent cucumber wastes and explore their valorization within the context of a biorefinery [24].

Herein, vacuum microwave hydro-distillation (VMHD) as an energy-efficient and rapid technique for isolating essential oils and residues from cucumber is explored and compared with respect to the conventional hydro-distillation method (HD). Within the context of a zero-waste biorefinery, extractive-free residues are a potential source of cellulose and hemicellulose. The former can be considered as a potential source of sugars and platform molecules, whilst the latter can be considered as a potential source of fibers. This may be the first time to characterize the cucumber residues after VMHD.

## 2. Materials and Methods

Cucumbers grown in Spain were purchased from Nisa (York, UK). The moisture content of cucumbers was determined (5 replicates: 96% $\pm$ 0.5%) by drying them in an oven at 105 °C until a constant weight was achieved. The unpeeled cucumbers were thoroughly washed with de-ionized water, chopped into chunks, mixed with deionized water (1 (wt):4 (wt) cucumber:water), and milled with a Retsch Knife Mill for 3 min at 3000 rpm to afford a slurry.

### 2.1. Extraction via Hydro-Distillation (HD)

The cucumber slurry was subjected to conventional hydro-distillation at 100 °C for 3 h until no more oil was obtained. The extraction yield was calculated via GC. The oils were isolated to afford an extractive yield of 3.2%.

### 2.2. Extraction via Vacuum Microwave Hydro-Distillation (VMHD)

Cucumber slurry was subjected to VMHD using a Milestone Roto SYNTH microwave with a 4 L Pyrex reactor vessel connected to a system an external condensers and vacuum (PC 3012 NT VARIO, Vacuubrand) as depicted in Figure 1.

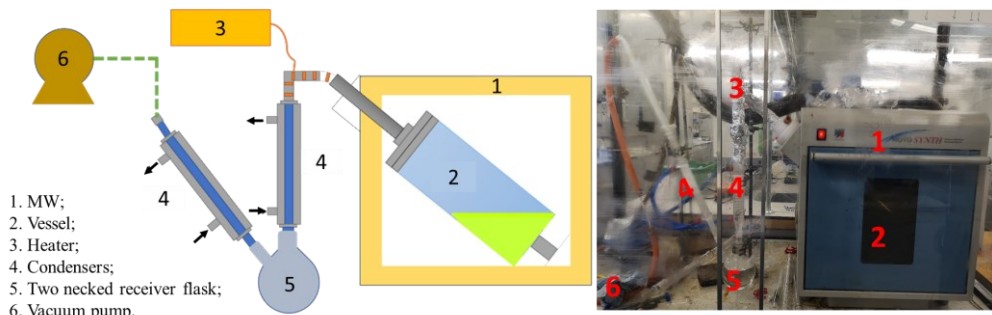

**Figure 1.** Vacuum microwave hydro-distillation set-up: 1. 45° rotative Milestone Roto SYNTH MW; 2. MW Vessel; 3. heater; 4. condensers; 5. two-necked receiver flask; 6. vacuum pump.

A two-step irradiation process of the experiment was applied (see Section 3.2.2 for details): (i) the cucumber slurry was rapidly heated to boiling (1000 W power, 2 min 20 s); (ii) boiling was then maintained using stable irradiation power (500, 750, 1000 W) under constant vacuum (100, 200, 300 mbar). Microwave irradiation was stopped as soon as most of the water was removed from the slurry and the time was recorded accordingly. The aqueous–oily mixture (condensate) in the receiving flask was transferred to a separating funnel. The oil was extracted into dichloromethane (10 mL), dried ($MgSO_4$), and concentrated in vacuo to afford the desired VMHD extractive. The extraction yield was calculated via GC. The overview of the isolation procedure is shown in ESI S1.

### 2.3. Chemical Analysis

#### 2.3.1. Gas Chromatography–Mass Spectrometry Analysis

The resulting oils were characterized by the HP01 method on Agilent 7890A connected to a single quadrupole Agilent 5975C MS equipment: the initial oven temperature was 40 °C and held for 2 min, ramped at a rate of 4 °C min$^{-1}$; when the final temperature of 250 °C was reached, it was held for 10 min; the split ratio was 5:1, and injector temperature was 250 °C; the column was a fused-silica capillary DB-5HT (30 m × 0.25 mm × 10 μm) with a carrier gas of helium at 99.99% purity. The identification of the detected compounds was confirmed by the NIST database or Wiley 7 libraries. The typical chromatograms of cucumber essential oil from VMHD and HD are presented in ESI S2 and S3, respectively.

#### 2.3.2. $^{13}$C Solid-State Nuclear Magnetic Resonance

The solid NMR spectra were obtained on a JEOL 400S instrument, the $^{13}$C frequency was 10 k Hz, spectra were collected by using a 4 mm CP-MAS probe with a sample spinning rate of 10,000 Hz, and the scan number was 1024.

#### 2.3.3. Thermogravimetric Analysis

The thermogravimetric properties of the samples were recorded using a PL Thermal Science STA 625. The samples were weighed accurately in an alumina pan and analyzed in a nitrogen atmosphere from 20 to 625 °C at a heating rate of 10 °C min$^{-1}$; an empty alumina pan was as a reference.

#### 2.3.4. Attenuated Total Reflectance Infrared Spectroscopy

ATR-IR was recorded on a Perkin Elmer FT-IR Spectrometer using Spectrum software, 64 background scans and 32 sample scans were recorded for each sample, and the scan range was from 650 to 4000 cm$^{-1}$.

#### 2.3.5. Elemental Analysis

Elemental analysis was performed on an Exeter Analytical CE-440 analyzer in conjunction with a Sartorius SE2 analytical balance. Samples were combusted at 975 °C in an

oxygen atmosphere, and various thermal conductivity detectors analyzed the combustion of products. Analysis was conducted 3 times.

### 2.3.6. Scanning Electron Microscopy

The morphology of the sample was imaged using a JSM-6490LV from JEOL; to avoid disturbance, the samples were kept under liquid nitrogen conditions and coated with gold.

## 3. Results and Discussion

### 3.1. Biomass Characterization

To gain a deeper understanding of cucumbers, the property of raw materials should be explored. Herein, the composition of oven-dried cucumber is investigated, providing support for the following research.

Thermogravimetric Analysis

The thermogram of dried cucumber slurry is shown in Figure 2. The residual mass at 625 °C is around 30.3%. The mass loss from 25 to 100 °C is associated with moisture and volatiles (about 5%). Further losses (approx. 25%) are noted from 150 to 200 °C which may be attributed to small molecule volatilization (essential oils) and hemicellulose decomposition. The mass loss at about 310 °C is characteristic of cellulose decomposition (approx. 30%), and the decomposition at 375 °C may be due to the lignin (approx. 8%).

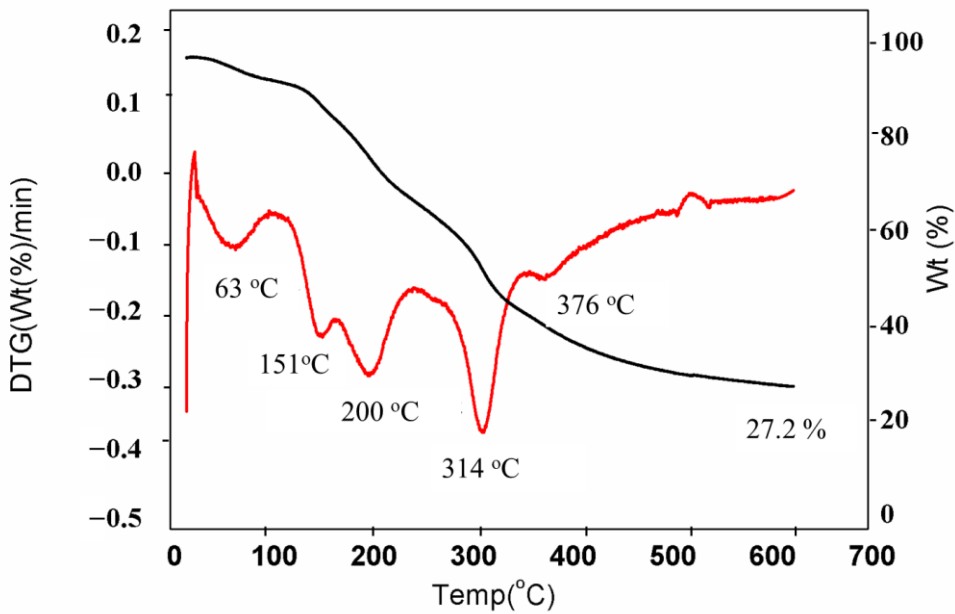

**Figure 2.** Thermogravimetric analysis of oven-dried cucumber.

### 3.2. Vacuum Microwave Hydro-Distillation of Essential Oil

3.2.1. Identification of Main Fraction Compounds of the Essential Oil from Cucumber Slurry by Vacuum Microwave Hydro-Distillation

As shown in Table 1, a maximum of 17 compounds were identified. The essential oils are found to be rich in unsaturated aldehydes (2,6-nonadienal, 2-nonenal) aldehydes (furfural, 7,10,13-hexadecatrienal), and alcohols (1,16-hexadecanediol). 7,10,13-Hexadecatrienal was the most plentiful component, followed by 2- nonenal and 1,16-hexadecanediol [25].

**Table 1.** Identification of main fraction compounds of the essential oil samples extracted from cucumbers by VMHD.

| # | Retention Time (min) | Identified Compound | Formula | Molecular Weight |
|---|---|---|---|---|
| 1 | 4.89 | Furfural | $C_5H_4O_2$ | 96.08 |
| 2 | 11.6 | 2,6-nonadienal | $C_9H_{14}O$ | 138 |
| 3 | 12.2 | 1-pentadecyne | $C_{15}H_{28}$ | 208 |
| 4 | 13.0 | 2,6-nonadienal | $C_9H_{14}O$ | 138 |
| 5 | 13.0 | 2-nonenal | $C_9H_{16}O$ | 140 |
| 6 | 14.2 | 2-undecene | $C_{13}H_{26}O_2$ | 214 |
| 7 | 16.4 | nonanoic acid | $C_9H_{18}O_2$ | 158 |
| 8 | 17.6 | methyl 9-oxo-10-undecenoate | $C_{12}H_{22}O_2$ | 198 |
| 9 | 18.8 | 1-pentadecyne | $C_{15}H_{28}$ | 208 |
| 10 | 19.8 | 1,16-hexadecanediol | $C_{16}H_{34}O_2$ | 258 |
| 11 | 21.7 | 7,10,13-hexadecatrienal | $C_{16}H_{26}O$ | 234 |
| 12 | 23.4 | 9,12,15-octadecatrien-1-ol, (9Z,12Z,15Z)- | $C_{18}H_{32}O$ | 264 |

### 3.2.2. Vacuum Microwave Hydro-Distillation Optimization

To explore the relationship between various parameters (vacuum, power) and irradiation time costs together with their corresponding essential oil yields, nine tests were carried out. From Table 2, it can be concluded that at the same vacuum, a higher power reduced the extraction time. Meanwhile, at fixed microwave power, a higher vacuum resulted in a shorter extraction time. The highest essential oil yield (0.49 wt %) was obtained at 500 W and 300 mbar after 54 min 10 s, and the lowest oil yield (0.07%) was obtained at 750 W and 100 mbar after 25 min. The results show that the amount of essential oil is probably not influenced by the power and/or vacuum. For example, increasing microwave power from 500 to 1000 W had no significant effect on essential oil yield [26]. However, a stronger vacuum will result in a lower yield. Overall, in this study, the pressure and power probably can only affect the boiling point of cucumber slurry and extraction time, and the extraction time played a crucial role in the yield of essential oils.

**Table 2.** Influence of experimental parameters on oil yield.

| Exp # | Power (W) | Pressure (mbar) | Time | Power (W) | Pressure (mbar) | Temp (°C) | Time (min) | Yield (%, Dry Basis) |
|---|---|---|---|---|---|---|---|---|
| | | Step 1 | | | Step 2 | | | |
| 1 | 1000 | 1024 | 2 min 20 s | 500 | 100 | 60 | 34 min 20 s | 0.22 |
| 2 | 1000 | 1024 | 2 min 20 s | 750 | 100 | 60 | 25 min | 0.07 |
| 3 | 1000 | 1024 | 2 min 20 s | 1000 | 100 | 60 | 20 min | 0.29 |
| 4 | 1000 | 1024 | 2 min 20 s | 500 | 200 | 67 | 47 min | 0.12 |
| 5 | 1000 | 1024 | 2 min 20 s | 750 | 200 | 67 | 26 min 10 s | 0.09 |
| 6 | 1000 | 1024 | 2 min 20 s | 1000 | 200 | 67 | 21 min | 0.13 |
| 7 | 1000 | 1024 | 2 min 20 s | 500 | 300 | 73 | 54 min 10 s | 0.49 |
| 8 | 1000 | 1024 | 2 min 20 s | 750 | 300 | 73 | 34 min 40 s | 0.20 |
| 9 | 1000 | 1024 | 2 min 20 s | 1000 | 300 | 73 | 21 min 30 s | 0.25 |

### 3.2.3. Vacuum Microwave Hydro-Distillation versus HD

Comparing the essential oil samples obtained from HD and VMHD, no significant identified compound difference was observed in this study (ESI S2 and S3). However, compared to the conventional distillation method (around 3 h), VMHD reduces extraction time significantly (less than 1 h). This can be attributed to the microwave mechanism of

heat penetration into the plant material: the water molecules inside the cucumber absorb the irradiation from the microwave, and then the microwave energy is directly converted into heat, resulting in a sudden temperature increase inside the materials. Afterward, the plant cells are under severe thermal stress and localized in relatively high pressures, which exceed the capacity of cells; this destroys the structure of the material more rapidly and efficiently than conventional extraction methods such as HD in which the heat transfers from the outside to the inside of the material [27].

### 3.3. Residue Analysis

The solid residues after isolation are normally regarded as biopolymers such as polyphenols or insoluble cellulosic materials, and these materials have the potential to be used as energy or bio-compound derivatives. Thus, it is necessary to identify the components and characterize the features of the waste to evaluate the potential value of these residues. This investigation aimed to assess the potential utilization of cucumber residues after the VMHD process, and the effects of treatment with variable parameters (vacuum, power) on residues were investigated.

### 3.3.1. $^{13}$C CP/MAS Solid-State NMR

The stacked $^{13}$C CP/MAS solid-state NMR spectra of cucumber residues after extraction (either VMHD or HD) are presented in Figure 3. The signals at around 175–180 ppm derive from the carbonyl group (C=O) present in hemicellulosic and pectinaceous matter. The carbon signals of cellulose ($C_1$ to $C_6$) were displayed, ranging from 110 ppm to 60 ppm. Furthermore, no obvious changes between the cellulosic amorphous region in $C_4$ (84 ppm) and $C_6$ (62 ppm) and the crystalline region in $C_4$ (89 ppm) and $C_6$ (65 ppm) were observed; this suggests that the amorphous cellulose had not been hydrolyzed to monosaccharides during VMHD and HD procedures. The carbon signals at around 100 ppm correspond to glycosidic bond carbons $C_1'$ [28]; additionally, the carbon signals ($C_4'$, $C_{2,3,5}'$, $C_6'$) of pectin ranging from 60 ppm to 80 ppm were probably overlapped by the signals of cellulose ($C_4$, $C_{2,3,5}$, and $C_6$). The peaks at around 54 ppm may be attributed to methyl carbons of the methyl ester pectinates in the cucumber. The results indicate that after VMHD or traditional HD treatment, the cucumber residues were mainly polysaccharides (e.g., cellulose, hemicellulose, and pectin), and no significant component changes in the residue were observed even with different process conditions (power, vacuum).

### 3.3.2. ATR-IR

The ATR-IR spectra of cucumber residues with various VMHD conditions (vacuum, power) and hydro-distillation treatment are summarized in Figure 4. The vibration at around 3300 cm$^{-1}$ could be related to the O-H stretching which refers to the bonded water existing in the residue. The bands at 2910 cm$^{-1}$ may refer to the C-H stretch from cellulose/hemicellulose. The absorption bands at about 2860 cm$^{-1}$ correspond to C-H asymmetric and symmetric vibration in saturated $CH_2$. The presence of hemicellulose and/or pectinaceous matter is proved by C=O stretching at 1732 cm$^{-1}$ and 1602 cm$^{-1}$ indicative of methyl ester carbonyl and free carboxyl.

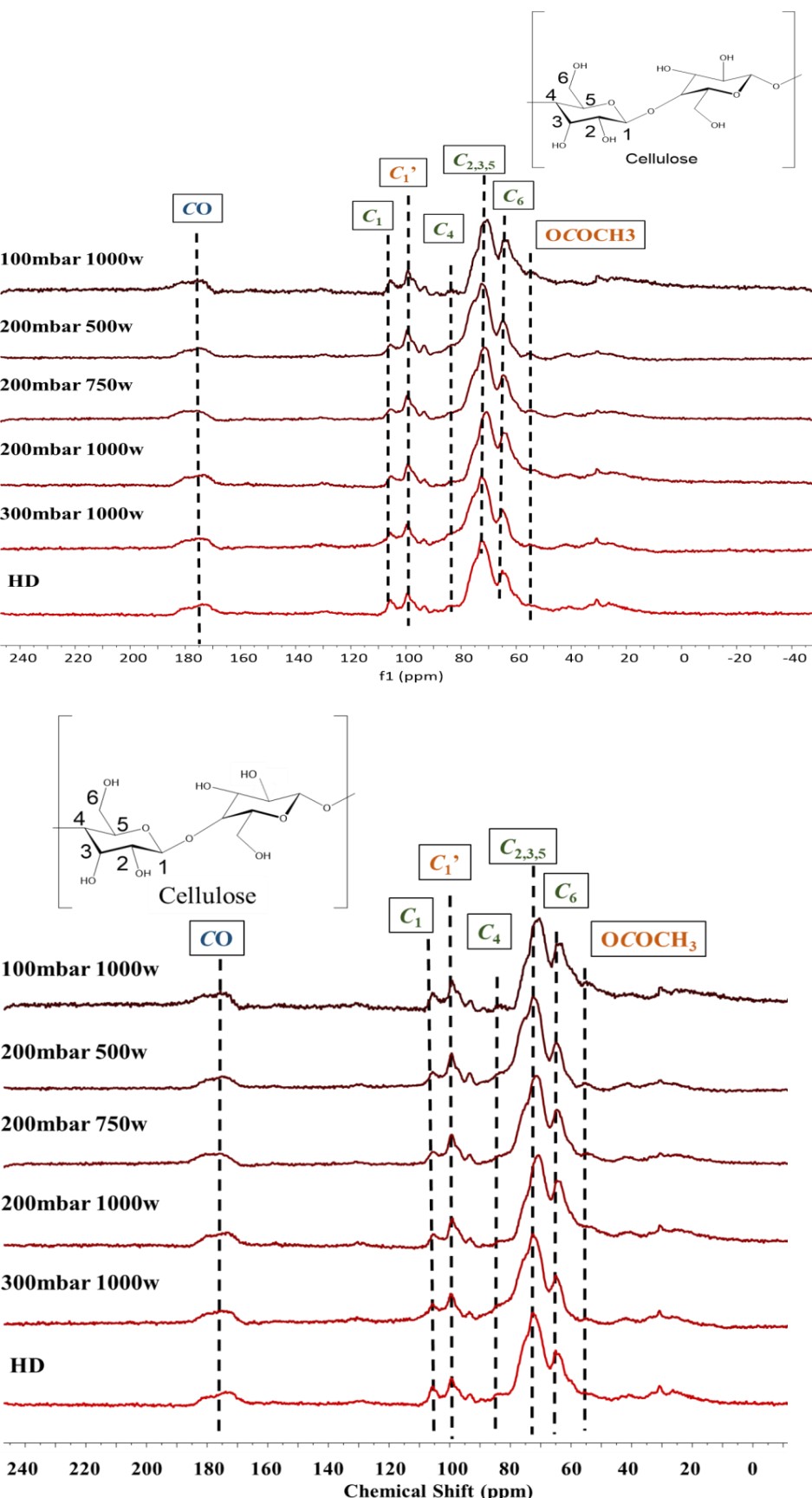

**Figure 3.** $^{13}$C CP/MAS NMR spectra of cucumber residues obtained from HD and VMHDF.

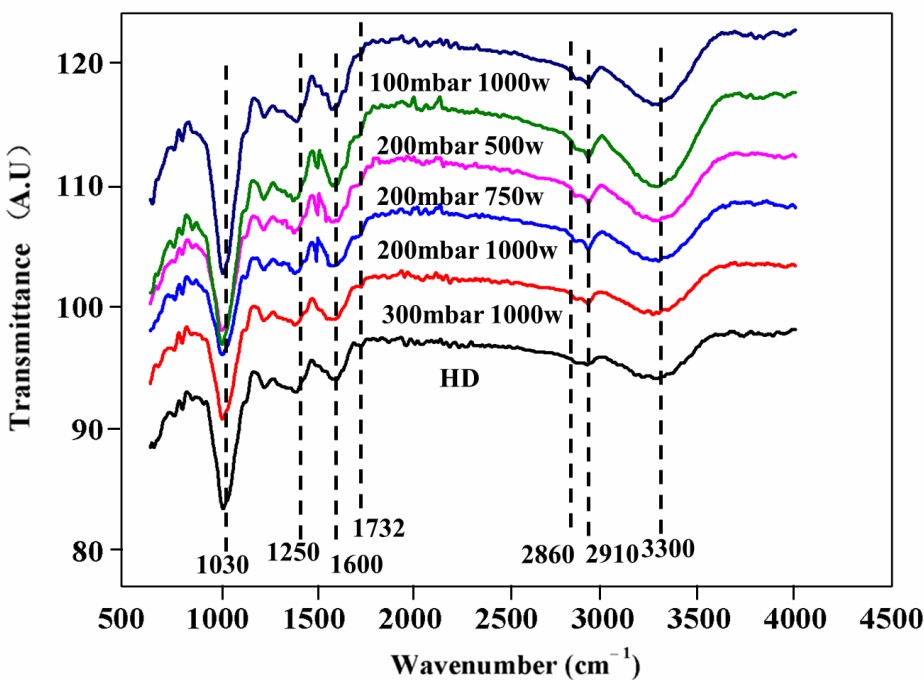

**Figure 4.** ATR-IR of cucumber residues obtained from HD and VMHD.

### 3.3.3. Elemental Analysis

The percentage composition of carbon, hydrogen, and nitrogen for oven-dried cucumbers and their residues obtained from VMHD (500 W, 300 mbar) are shown in Figure 5. As can be seen from the figure, the oven-dried cucumber displays higher carbon content (44.84%), while the residue after VMHD shows a carbon content of 39.52%; this may be due to the volatile compounds which are rich in carbon being carried away with water vapor during VMHD treatment, resulting in a relatively lower carbon content; meanwhile, no noticeable differences in nitrogen (3 to 2.3%) or hydrogen content (5.1 to 6.2%) were found between the samples.

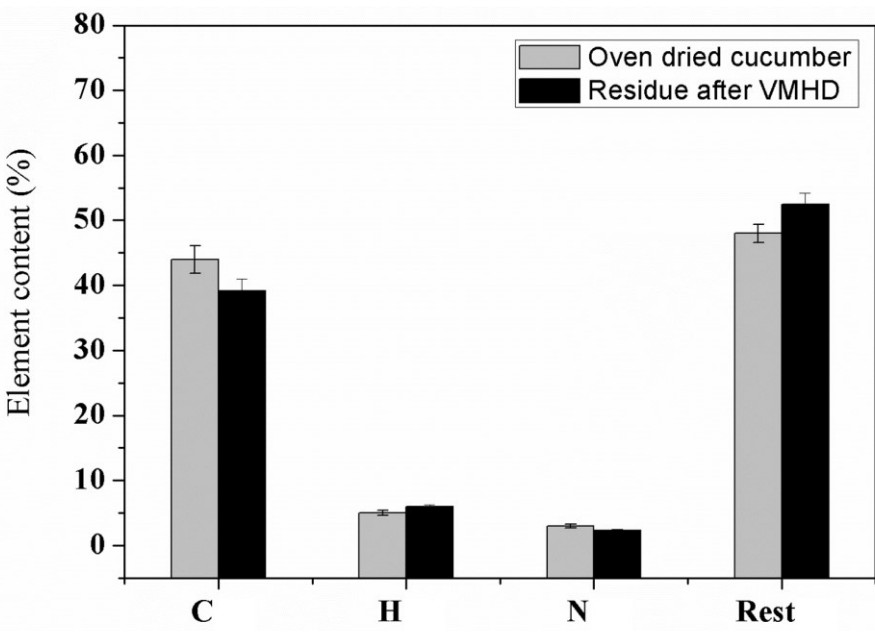

**Figure 5.** CHN analysis of cucumbers and their residues obtained from VMHD (500 W, 300 mbar).

### 3.3.4. Surface Morphology Changes of Cucumber Residue after VMHD and HD

To investigate the effect of extraction methods and different factors on the surface morphology of cucumber residues, SEM images of cucumber residues obtained from VMHD and HD are shown in Figure 6. It can be seen that an irradiation power of 1000 W gives visible disruption on the external surface (see Figure 6a–c, highlighted by red circles). At the same vacuum pressure (200 mbar), the disruptions are also observed on the surface morphology when the power ranges from 500 to 750 W (see Figure 6d,e). Meanwhile, with hydro-distillation treatment (Figure 6f), the sample displays a similar morphology feature to VMHD: apparent rupture is also observed. However, the surfaces are smoother; this may be due to the difference in the rate of energy transfer between the two heating methods: the heat transfer is mainly performed by conduction and convection in HD treatment, and before the heat energy reach the plant surface, the solvent has been heated; this leads to a surface rupture of the plant during the process.

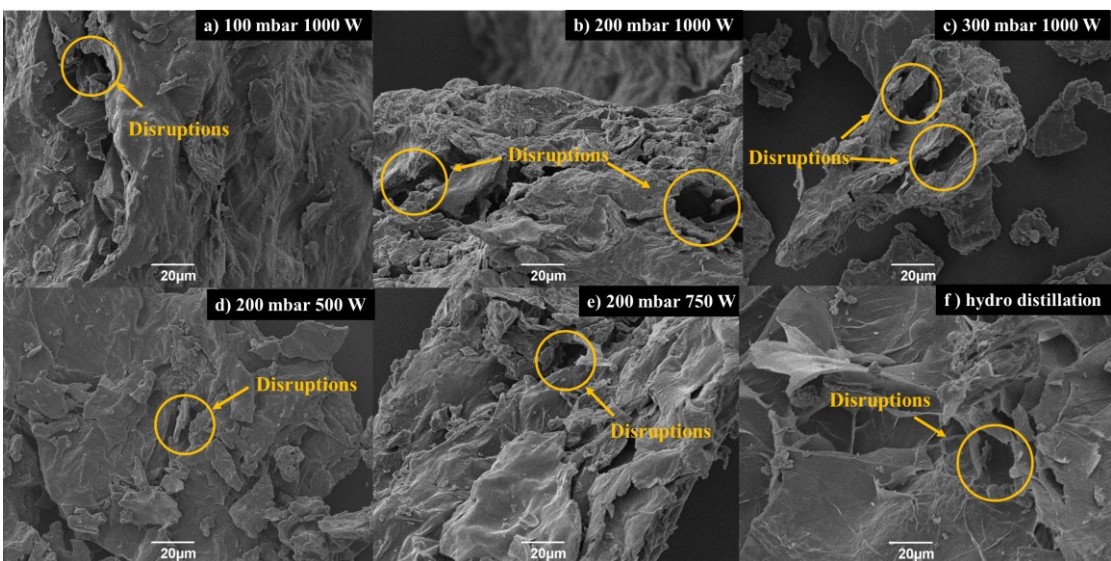

**Figure 6.** Morphological features of cucumber residues (scale bar = 20 μm) produced using various experimental conditions during VMHD (pressure, power): (**a**) 100 mbar 1000 W, (**b**) 200 m 1000 W, (**c**) 300 m 1000 W; (**d**) 200 mbar 500 W, (**e**) 200 mbar 750 W, and (**f**) hydro-distillation. The disruptions are highlighted in red circles.

### 4. Conclusions and Prospects

Spent cucumber in the form of a cucumber slurry has the potential to be useful renewable feedstock for a zero-waste biorefinery. The utilization of spent cucumber is in line with Green Chemistry Principle 7 and UN Sustainable Development Goal 12 (Responsible Production and Consumption), which invokes waste re-utilization, doing more with less, and resource circularity. Spent cucumber waste is a potential source of biochemicals (essential oils) and residual cellulose matter which could be converted into useful platform molecules and biomaterials which can be used in various applications including food, membranes, and textiles. However, due to the restriction of equipment, the comparison of essential oil from VMHD and HD has not been studied; meanwhile, the further utilization of the cellulosic materials after VMHD should be implanted, for example, in the form of films and/or gels made of cucumber residues. In the future, a total comparison between HD and VMHD should be implemented, and the practical application of cucumber residues after VMHD needs to be explored. Microwave technology is a suitable technology for the extraction of oils; it is rapid and energy-efficient.

**Author Contributions:** Conceptualization, J.F. and Y.G.; methodology, J.F. and H.K.B.; writing—original draft preparation, Y.G.; writing—review and editing, Y.G. and A.S.M.; supervision, J.F. and Y.G.; project administration, J.F. All authors have read and agreed to the published version of the manuscript.

**Funding:** This work is financially supported by the EPSRC (grant number EP/R51181X/1).

**Data Availability Statement:** Data available in a publicly accessible repository.

**Acknowledgments:** The authors would also like to thank Vitaliy L. Budarin for help with the experiment design.

**Conflicts of Interest:** The authors declare no conflict of interest.

## Abbreviations

| | |
|---|---|
| VMHD | Vacuum microwave hydro-distillation |
| HD | Hydro-distillation |
| MAHD | Microwave-assisted hydro-distillation |
| MASD | Microwave-accelerated steam distillation |
| MHG | Microwave hydro-diffusion and gravity |
| MSD | Microwave steam distillation |
| SFME | Solvent-free microwave extraction |
| GC-MS | Gas chromatography–mass spectrometry |
| $^{13}$C Solid-State NMR | $^{13}$C solid-state nuclear magnetic resonance |
| TGA | Thermogravimetric analysis |
| ATR-IR | Attenuated total reflectance infrared |
| SEM | Scanning electron microscopy |

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
