# Peer review of "Exploration of Cucumber Waste as a Potential Biorefinery Feedstock"

_processes, doi:10.3390/pr10122694_

Round 1

Reviewer 1 Report

Process
Manuscript ID: processes-1992934
Exploration of cucumber waste as a potential biorefinery feed-2 stock

The manuscript is of general interest. The following comments should help further improve the quality of the work:

1-Please add two more keywords (up to 6 is allowed). Metadata, including keywords, are important in terms of the searchability of the manuscript if published.
2-Please avoid using abbreviations as keywords.
3-Please include a Table of Abbreviations/Nomenclatures.
4-The manuscript should be improved in terms of the usage of English. Authors are advised to get their manuscript edited by a native English speaker or by a Professional English Editing Service.
5-Scientific names (
Cucumis sativus) should be italicized. Please note that “L” in “Cucumis sativus L” should not be italicized.
6-The manuscript should also be improved in terms of the application of punctuation marks.
7-Authors have reported an interesting pathway for valorizing biomass. This also emphasizes that the importance of biomass in the current and future bioeconomy is generally beyond the general perception. In a recent work entitled “Beyond conventional biomass valorisation: pyrolysis-derived products for biomedical applications” this has also been emphasized. Authors can consider including the mentioned work to further elaborate on this.
8-
The novelty of the present work against the existing literature in this domain should be more effectively highlighted.
9-Please do not repeat units where not needed, like in “(100 mbar, 200 mbar, 300 mbar)” which should be “(100, 200, and 300 mbar)”.
10-Using t
oo long paragraphs should be avoided.
11-Please avoid using Abbreviations in headings and sub-heading throughout the manuscript.
12-Please number all headings and sub-headings properly.
13-Figure 1 should be graphically enhanced.
14-Using headings after headings with no explanation should be avoided. See 3, 3.1, and 3.1.1, for instance.
15-There should be space between terms and units in parentheses, like in the labels placed on the X axis of Figure 2.
16-The formula presented in Table 1, need to be revised. Numbers should be subscripted. Please scan the manuscript for such errors and remove them.
17-Please make sure all the units will be presented in compliance with the SI System. For instance, please use "min" for "minutes" and not “m”, etc. This comment applied to the units used in Figures/Tables too.
18-Please reduce the significant figures of the reported data to three. Here`s an example of significant figures (sig figs):
- 10082 (5 sig figs)
- 70,000 (1 sig fig)
- 0.0025 (2 sig figs)
- 0.000309 (3 sig figs)
- 50010.000 (8 sig figs)
- 0.0040030 (5 sig figs)”
19-Please make sure consistency will be observed in presenting units, either use "/" or "-1" style. This comment applied to the units used in Figures/Tables too.
20-The texts (in red) presented in Figure 6 do not read. Please improve this figure.
21-Using advanced sustainability assessment tools such as life cycle assessment and exergy analysis, as explained in a recent work “The role of sustainability assessment tools in realizing bioenergy and bioproduct systems”, is expected to offer some interesting insights into the sustainability aspects of the developed biorefinery platform. Authors can consider including the mentioned work to highlight the importance of such assessments and to direct future studies.
22-The limitations of the study should also be explained.
23-The practical implication of the present study should be included and discussed as well.
24-Please change "4. Conclusions" to "4. Conclusions and prospects". Accordingly, please elaborate on the future research needs in this domain.

25-It is advisable to add DOIs for the references.

Author Response

Thank you very much for your careful review and constructive suggestions with regard to our manuscript. Meanwhile, I am also very grateful to you for the specific modification strategy.

1-Please add two more keywords (up to 6 is allowed). Metadata, including keywords, are important in terms of the searchability of the manuscript if published.

Response: New keywords: hydro-distillation and biorefinery were added.

2-Please avoid using abbreviations as keywords.

Response: The abbreviation was replaced by the full name.

3-Please include a Table of Abbreviations/Nomenclatures.

Response: The table of Abbreviations/Nomenclatures were added.

Abbreviations

VMHD

Vacuum microwave hydro-distillation

HD

Hydro-distillation

MAHD

Microwave-assisted hydro-distillation

MASD

Microwave-accelerated steam distillation

MHG

Microwave hydro-diffusion and gravity

MSD

Microwave steam distillation

SFME

Solvent-free microwave extraction

GC-MS

Gas Chromatography-Mass Spectrometer

13C Solid-State  NMR

13C Solid-State Nuclear Magnetic Resonance

TGA

Thermogravimetric analysis

ATR-IR

Attenuated Total Reflectance Infrared

SEM

Scanning electron microscopy

4-The manuscript should be improved in terms of the usage of English. Authors are advised to get their manuscript edited by a native English speaker or by a Professional English Editing Service.

Response: The manuscript has been improved by a native English speaker.

5-Scientific names (Cucumis sativus) should be italicized. Please note that “L” in “Cucumis sativus L” should not be italicized.

Response: Scientific names (Cucumis sativus) have been italicized

6-The manuscript should also be improved in terms of the application of punctuation marks.

Response: The punctuation marks were thoroughly checked and improved.

7-Authors have reported an interesting pathway for valorizing biomass. This also emphasizes that the importance of biomass in the current and future bioeconomy is generally beyond the general perception. In a recent work entitled “Beyond conventional biomass valorisation: pyrolysis-derived products for biomedical applications” this has also been emphasized. Authors can consider including the mentioned work to further elaborate on this.

Response: The work has been cited.

8-The novelty of the present work against the existing literature in this domain should be more effectively highlighted.

Response: The novelty of the present work had been highlighted in line 78-79: This may be the first time to characterise the cucumber residue after VMHD.

9-Please do not repeat units where not needed, like in “(100 mbar, 200 mbar, 300 mbar)” which should be “(100, 200, and 300 mbar)”.

Response: The units where not needed were deleted.

10-Using too long paragraphs should be avoided.

Response: Paragraph 2 has been separated.

11-Please avoid using Abbreviations in headings and sub-heading throughout the manuscript.

Response: The abbreviations in headings and sub-heading were replaced by the full name.

12-Please number all headings and sub-headings properly.

Response: All headings and sub-headings were numbered.

13-Figure 1 should be graphically enhanced.

Response: Figure 1 has been improved.

14-Using headings after headings with no explanation should be avoided. See 3, 3.1, and 3.1.1, for instance.

Response: The explanations were added before 3, 3.1, and 3.1.1.

To gain a deeper understanding of cucumbers, the property of raw materials should be explored. Herein, the composition of oven-dried cucumber was investigated, providing support for the following research.

3.1.1

The solid residues after isolation are normally regarded as biopolymers like polyphenols or insoluble cellulosic materials, and these materials have the potential to be used as energy or bio-compounds derivatives. Thus, it is necessary to identify the components and characterize the features of the waste to evaluate the potential value of these residues. This investigation aimed to assess the potential utilization of cucumber residues after the VMHD process, and the effects of treatment with variable parameters (vacuum, power) on residues were investigated.

3, 3.1

15-There should be space between terms and units in parentheses, like in the labels placed on the X axis of Figure 2.

Response: The space in the labels placed on the X axis of Figure 2 has been added.

16-The formula presented in Table 1, need to be revised. Numbers should be subscripted. Please scan the manuscript for such errors and remove them.

Response: All numbers in Table 1 has been subscripted and such errors had been corrected.

17-Please make sure all the units will be presented in compliance with the SI System. For instance, please use "min" for "minutes" and not “m”, etc. This comment applied to the units used in Figures/Tables too.

Response: The “m” was replaced by "min".

18-Please reduce the significant figures of the reported data to three. Here`s an example of significant figures (sig figs):

- 10082 (5 sig figs)

- 70,000 (1 sig fig)

- 0.0025 (2 sig figs)

- 0.000309 (3 sig figs)

- 50010.000 (8 sig figs)

- 0.0040030 (5 sig figs)”

Response: The significant figures in Table 1 had been reduced to three.

19-Please make sure consistency will be observed in presenting units, either use "/" or "-1" style. This comment applied to the units used in Figures/Tables too.

Response: The units are now in line.

20-The texts (in red) presented in Figure 6 do not read. Please improve this figure.

Response: Figure 6 has been improved.

21-Using advanced sustainability assessment tools such as life cycle assessment and exergy analysis, as explained in a recent work “The role of sustainability assessment tools in realizing bioenergy and bioproduct systems”, is expected to offer some interesting insights into the sustainability aspects of the developed biorefinery platform. Authors can consider including the mentioned work to highlight the importance of such assessments and to direct future studies.

Response: Thank you, such assessments would be considered in the future studies.

22-The limitations of the study should also be explained.

Response: The limitations of the study had been explained in Conclusions and prospects:

However, due to the restriction of equipment, the comparison of essential oil from VMHD and HD has not been studied; meanwhile, the further utilization of the cellulosic materials after VMHD should be implanted, for example, films and/or gels made of cucumber residues.

23-The practical implication of the present study should be included and discussed as well.

Response: The practical implication was added: Spent cucumber waste is a potential source of biochemicals (essential oils) and residual cellulose matter which could be converted in to useful platform molecules and biomaterials which can be used in various applications including food, membranes or textiles.

24-Please change "4. Conclusions" to "4. Conclusions and prospects". Accordingly, please elaborate on the future research needs in this domain.

Response: "4. Conclusions" had been changed to "4. Conclusions and prospects".

The future research was added: In future, a total comparison between HD and VMHD should be Implemented, and the practical application made of cucumber residues after VMHD need to be explored.

25-It is advisable to add DOIs for the references.

Response: DOIs were added.

Reviewer 2 Report

The conventional hydro-distillation temperature (100oC) is not suitable for the test solution
This research has not presented the efficiency of each steps in the oil extraction process.
The author has not presented the method of calculating oil yield in the experiments and the error in these data

Author Response

Thank you very much for your careful review and constructive suggestions with regard to our manuscript. Meanwhile, I am also very grateful to you for the specific modification strategy.

Manuscript ID: processes-1992934

Comments and Suggestions for Authors

The conventional hydro-distillation temperature (100 oC) is not suitable for the test solution

Response: We agree that conventional hydro-distillation temperature (100 oC) may not be ideal control but given that steam distillation is near 100 oC, then the former is a suitable proxy.

research has not presented the efficiency of each steps in the oil extraction process.

Response: Thanks for the comment. We agree that it is important to present the efficiency of each steps in the oil extraction process. As this paper is mainly focused on the re-utilization of the extractive-free residues, we didn't consider this important step, but in the future we won’t forget this.

The author has not presented the method of calculating oil yield in the experiments and the error in these data.

Response:

The method: The extraction yield was calculated via GC and were added to line 89.

We appreciate the reviewer’s insightful suggestion and agree that it would be useful. As this paper is mainly focused on the re-utilization of the extractive-free residues, the oil yield was calculated only once, but this would be avoided in the future research.

Reviewer 3 Report

There are many typographical errors, manuscript to be thoroughly checked. Proper use of subscript and superscripts in degree centigrade, chemical formulas are missing.

The name of species must be in italics, correct throughout the manuscript.

Is there any unidentified peaks?

There is a mismatch between abstract and conclusion, there is a lack of comparison between HD and VMHD in totality, this may be expressed in detail.

Author Response

Thank you very much for your careful review and constructive suggestions with regard to our manuscript. Meanwhile, I am also very grateful to you for the specific modification strategy.

Manuscript ID: processes-1992934

Comments and Suggestions for Authors

There are many typographical errors, manuscript to be thoroughly checked. Proper use of subscript and superscripts in degree centigrade, chemical formulas are missing.

Response: The manuscript was thoroughly checked: including subscript and superscripts in degree centigrade, and missing chemical formulas.

The name of species must be in italics, correct throughout the manuscript.

Response: Scientific names (Cucumis sativus) had been italicized

Is there any unidentified peaks?

Response: Yes,  the peaks were not identified in ESI S2 (6.18,9.12, 20.8, 29.0 etc.)due to the restriction of equipment.

There is a mismatch between abstract and conclusion, there is a lack of comparison between HD and VMHD in totality, this may be expressed in detail.

Response: Thanks for the comment. As this paper is mainly focused on the re-utilization of the extractive-free residues, a detailed comparison between HD and VMHD were not implemented. However, this would avoid in the future research.

Round 2

Reviewer 1 Report

The manuscript can be accepted for publication.

Author Response

All figures were carefully improved.

Reviewer 3 Report

The raised concern have been addressed

Author Response

All figures were carefully improved.